# Impacts of Ecological Shading by Roadside Trees on Tea Foliar Nutritional and Bioactive Components, Community Diversity of Insects and Soil Microbes in Tea Plantation

**DOI:** 10.3390/biology11121800

**Published:** 2022-12-12

**Authors:** Yan Zou, Yanni Zhong, Han Yu, Sabin Saurav Pokharel, Wanping Fang, Fajun Chen

**Affiliations:** 1Department of Entomology, College of Plant Protection, Nanjing Agricultural University, Nanjing 210095, China; 2Department of Forest Genetics and Breeding, College of Forestry, Nanjing Forestry University, Nanjing 210037, China; 3Department of Tea Science, College of Horticulture, Nanjing Agricultural University, Nanjing 210095, China

**Keywords:** ecological shading, roadside trees, foliar nutritional and bioactive components, population and community, insects and soil microbiome, ecological management in tea plantation

## Abstract

**Simple Summary:**

Tea is an important cash crop and is deeply loved by people around the world. The production of tea is affected by a variety of factors, and a landscape of roadside trees exists in many tea plantation ecosystems. What is the impact of this type of landscape on the production of tea? The management of plant diseases and insect pests in tea gardens, as well as the enhancement of tea quality, have been shown to be facilitated by appropriately shading and intercropping with high-stalk crops. We evaluated the impact of a roadside tree landscape on tea production, in terms of tea quality, pest occurrence, and soil microbial diversity. The results showed that planting roadside trees and providing shade for tea plants were beneficial to the improvement of tea quality and pest control in tea gardens, as well as soil quality. The research results can provide assistance for the ecological planning and construction of tea garden landscapes.

**Abstract:**

Roadside trees not only add aesthetic appeal to tea plantations, but also serve important ecological purposes for the shaded tea plants. In this study, we selected tea orchards with two access roads, from east to west (EW-road) and from south to north (SN-road), and the roadside trees formed three types of ecological shading of the adjoining tea plants; i.e., south shading (SS) by the roadside trees on the EW-road, and east shading and west shading (ES and WS) by the roadside trees on the SN-road. We studied the impacts of ecological shading by roadside trees on the tea plants, insects, and soil microbes in the tea plantation, by measuring the contents of soluble nutrients, bioactive compounds in the tea, and tea quality indices; and by investigating the population occurrence of key species of insects and calculating insect community indexes, while simultaneously assaying the soil microbiome. The results vividly demonstrated that the shading formed by roadside tree lines on the surrounding tea plantation (SS, ES, and WS) had adverse effects on the concentration of tea soluble sugars but enhanced the foliar contents of bioactive components and improved the overall tea quality, in contrast to the no-shading control tea plants. In addition, the roadside tree lines seemed to be beneficial for the tea plantation, as they reduced pest occurrence, and ES shading enhanced the microbial soil diversity in the rhizosphere of the tea plants.

## 1. Introduction

Roadside landscapes comprising roadside trees are well-known for their intrinsic natural beauty, rendering pleasantness with their mesmerizing greenery, which is of great importance. Roadside trees provide a variety of benefits, including shade, increased real estate values, improved road safety, stormwater runoff control, biological diversity support, and noise absorption [1,2,3]. Moreover, tree plants can absorb toxic gaseous pollutants and reduce airborne particulates, which benefits maintaining an eco-friendly environment, where the health risks from air pollution are alleviated [4,5,6]. The habitat encompassing shading by roadside trees is a very particular environmental condition, being quite different from the natural environment. In addition to their well-known functions as air purifiers and important natural embellishments, roadside trees also play a vital role in the development of sustainable ecology in the road–region ecosystem [7]. The roadside landscape harbors and directly impacts different native plant and animal species and acts as an ecological conduit, allowing species to move between vegetation patches in a highly fragmented landscape, preserving local flora and fauna species from extinction in agricultural production [8,9]. In modern agricultural production, the widespread adoption of monocropping and irrational use of chemical pesticides has negatively impacted the biodiversity of farmland ecosystems, including plants, animals, and soil microorganisms [10,11,12]. Generally, in areas away from cities, the ecological functions of roadside trees are more prominent [13,14]. The environment of roadside patches is exempt from the modern agricultural system, and the growth of different plants promotes the carbon cycle, increases soil microbial diversity, and reduces soil erosion [14,15,16]. Tall trees can provide a habitat for natural predatory species such as birds, which assist in controlling pests in farmland areas [17,18,19].

In many tea planting areas of China, tall trees are generally planted along the roadside. The intermittent shading of the tea cast by roadside trees creates shading stress with alterations in the sun angle and intensity of solar radiation [20]. In general, plants depend on light energy to carry out their basic functions, while shading stress directly impacts the photosynthetic rate though alterations in chlorophyll content and plant morphology [21]. Several studies have shown that proper shading can increase the foliar chlorophyll content of tea plants [21,22]. Tea, *Camellia sinensis* (L.), an evergreen perennial subtropical plant that prefers shade and can withstand humidity, is one of China’s most important cash crops. Shading immediately after the emergence of new shoots can improve the quality of the tea [23]. It has been reported that shading treatment in two consecutive production periods reduced the yield of late-emerging tea leaf buds and that timely removal of shading stress could reverse the yield reduction [24,25]. Shading stress also reduced the airflow and directly changed the soil humidity and temperature, and these changes altered the physicochemical properties of the soil and created microclimates that affected the structure and diversity of the soil microorganisms [21]. The shade from a roadside tree canopy is different from that provided by the shading nets widely seen in tea plantations, as it is rather intermittent and remains for longer.

In this study, tea leaves and buds were examined for their contents of essential soluble nutrients, bioactive metabolites, and quality indicators, under the influence of ecological shading by roadside trees, in order to explore the impacts of intermittent shading stress on the plant growth of the tea trees and tea quality. Furthermore, the ecological significance of roadside trees was assessed in terms of the occurrence of key insect pest species and the soil microbial community.

## 2. Materials and Methods

### 2.1. Experimental Site Description

The experimental tea plantation was situated at 31.72° N and 118.75° E in Hongqi Village, Jiangning District, Nanjing City, Jiangsu Province, China. The cultivar of tea tree was *Camellia sinensis cv*. Huangshan Zhong, and the species of roadside trees was camphor *Cinnamomum camphora* (L.). This tea orchard has been extensively used for commercial tea production for more than 10 years. At this experimental location, the average annual temperature was 16 °C, the average annual rainfall was 1073 mm, and the average days without frost were 224.

The tea was planted with a planting spacing of 0.5 m and a row spacing of 1.5 m in the east-west and south-north directions. The 4 m wide avenues connecting the tea orchard and camphor trees were planted with a 4 m row spacing and a 5 m plant spacing, both parallel to the tea planting direction. The roadside camphor trees were 1 m away from the adjoining tea plants, and the tree height was 6–8 m with a canopy of about 4–5 m. Based on the two-direction avenues from east to west (EW-road) and from south to north (SN-road), the roadside trees formed three types of ecological shading on the adjoining tea plants, i.e., south shading (SS) by roadside trees on the EW-road, and east shading and west shading by roadside trees on the SN-road. For the SS, and ES and WS shade treatments, the tea tree were planted in an east-west direction without shade (SSCK), and those in the north-south direction without shade (EWCK) were respectively set as controls. There were five ecological shade treatments (including: ES and WS vs. EWCK; SS vs. SSCK), and three survey regions (including 2 rows of tea plants for each survey region) were set as replicates for each treatment. Each repeated investigation included one 5 m-length row of tea plants and one Malaise trap for insect collection arranged in the middle of the two rows of tea plants. Two leaves and an active bud were collected on 15th August, 15th September, and 15th October of 2020 and 2021 for the following experiments, respectively. The detailed experimental field layout of the roadside tree ecological shading is shown in Figure 1.

During the entire experimental period in 2020 and 2021, photometers (Model: 1801C; Delixi Electric LTD, Leqing, China) were used to measure the light intensity on the tea plants in the five treatments (SS, ES, WS, SSCK, and EWCK) on ten sunny days in each sampling year, in order to research how roadside trees’ ecological shading affected the amount of light that tea plants received. The measurements were carried out at in real-time observations during different time periods, i.e., the average light intensity (6 h) on the canopy of tea plants in the ES and EWCK was measured from 6:00 a.m. to 12:00 a.m. (i.e., from sunrise to high noon time). While the WS and EWCK ecological shading was measured from 12:00 a.m. to 6:00 p.m. (i.e., high noon time to sundown), and the ecological shading treatments SS and SSCK were measured from 9:00 a.m. to 3:00 p.m. Two-way repeated-measures (ANOVAs) with the sampling year (2020 vs. 2021) and ecological shading as the main factors and sampling time as the repeated measure indicated that the amount of light falling onto the canopy of tea plants was significantly reduced by ecological shading (SS vs. SSCK: *F* = 3472.8, *p* < 0.001; ES and WS vs. EWCK: *F* = 1739.2, and 3322.8, *p* < 0.001), and there were significant differences in the light intensity between the two sampling years (2020 vs. 2021, SS and SSCK: *F* = 5.70, *p* < 0.02; ES and EWCK: *F* = 18.05, *p* < 0.001; WS and EWCK: *F* = 8.22, *p* = 0.02; Appendix A).

### 2.2. Determination of Foliar Nutritional and Bioactive Components of Plants

#### 2.2.1. Foliar Nutrient Contents

*Soluble sugar content.* The foliar content of soluble sugars in the tested tea leaves was determined using a test kit for plant soluble sugar content (No. A145-1-1; Nanjing Jiancheng Bioengineering Institute, Nanjing, China). The basis for this determination is that when concentrated sulfuric acid solution and sugar are combined, the resulting furfural or hydroxymethyl furfural can react with anthrone, and a change in the absorbance value is formed at 630 nm.

*Soluble protein content.* The foliar content of soluble sugars in the tea leaves that were tested was determined using a test kit for plant soluble sugar content (No. A145-1-1; Nanjing Jiancheng Bioengineering Institute, Nanjing, China). The anions in Coomassie brilliant blue can react with the amino termini in proteins to turn the solution blue when the dye is added to a sample that contains proteins. The amount of protein in the sample or standard solution can then be determined through measuring the absorbance.

*Free fatty acid content.* The foliar content of free fatty acids in the tea leaves tested was determined using a free fatty acids assay kit (A042-1-1; Nanjing Jiancheng Bioengineering Institute, Nanjing, China). Free fatty acids can form a copper salt of fatty acids by combining with copper ions and dissolving in chloroform, and a color change occurs.

#### 2.2.2. Foliar Bioactive Component Contents

*Polyphenols content.* The Folin–Ciocalteu colorimetric method was used to determine the phenolic content of the tea leaf extracts [26]. Folin–Ciocalteu reagent was used to oxidize tea leaf extracts, and sodium carbonate was used to stop the reaction. An MRX II Dynex plate reader was used to measure the absorbance at 760 nm after 90 min at 25 ℃ (Dynex Technologies, Inc., Chanilly, VA, USA). The absorbance values were compared with the standards for known concentrations of gallic acid.

*Caffeine content.* Using an HPLC-based technique, the amount of caffeine in the foliar tissues of tea plants was measured [27]. The tea samples were dried at 80 °C for 24 h, and then caffeine was extracted and purified.

*Theanine content.* An automatic amino acids analyzer (Hitachi L-8900, Tokyo, Japan) was used to measure the foliar contents of theanine in the test tea leaves. Theanine was measured by adding 5 mL of tea leaf extract to 5 mL of sulfo-salicylic acid and centrifuging the mixture at 13,000 rpm for 5 min, to facilitate the reaction. The mixture was filtered through a 0.20 µm nylon filter membrane and tested using an amino acid analyzer [28,29].

*Catechin content.* In a 70 °C water bath, the catechin in tea leaf samples was extracted using a 70% methanol aqueous solution. On a C18 column with a wavelength of 278 nm, catechins were determined using a gradient elution technique using an HPLC analysis method [30].

#### 2.2.3. Leaf Quality Index

Two leaf quality indices were used to measure the quality of green tea: the catechin quality index, and the phenol/ammonia (P/A) ratio. The results of the ISO international environmental test were used to quantify the leaf quality indices using the relative correction factor of catechins and caffeine [31].
The Catechin Quality Index = [EGCG (%) + ECG (%)]/EGC (%) × 100(1)
P/A Ratio = Polyphenol content/Amino acid content.(2)EGCG—epigallocatechin gallate; EGC—epigallocatechin; ECG—epicatechin gallate.

### 2.3. Insect Investigation

A total of 10 insect surveys were conducted in the tea orchard from July to October in 2020 and 2021, respectively. Random selection of 3 tea plants in each ecological shade treatment (including ES, WS, and SS) and the respective controls (EWCK and SSCK) was performed. Insects were counted in 10 day intervals, with identification of insect species. Moreover, the insects were collected using a Malaise trap set in each survey region for the three ecological shade treatments (ES, WS, and SS) and two controls (WECK and SSCK), and the species collected were also identified. In this experiment, there were two key species of insect pest, *Ricania cacaonis* Chou & Lu, 1977 and *Ectropis oblique* Prout, 1915, and their population abundance on tea plants was counted every 10 days and translated to the number of individuals per 100 tea plants, to study the effects of ecological shading by roadside trees on the population abundance of key pests in the tea plantation. Biodiversity indices were calculated based on the numbers of species and the numbers of insects. The formulae thus incorporated during the calculation were as follows:

Shannon–Wiener diversity index:(3)$$H=-\sum_{i=1}^{S}P_{i}\times \ln(P_{i})\;\;\;\;\;P_{i}=N_{i}/N$$

Pielou evenness index:(4)$$E=H/H_{\max}\;\;\;\;H_{\max}=\ln S$$

Margalef richness index:(5)D=(S−1)/lnN

Simpson dominance index:(6)$$C=\sum_{i=1}^{S}{\left(P_{i}\right)}^{2}\;\;\;\;P_{i}=N_{i}/N$$
*P*_i_: relative abundance of insect species i; *N*_i_: number of individuals for species i; *N*: the total number of individuals of all species in the community; *S*: the number of species in the community; *H*_max_: maximum species diversity index.

### 2.4. Composition and Diversity of Soil Microbial Community

On 2nd October 2021, one sampling site was randomly selected in each survey region of the five treatments (i.e., 3 sampling sites per treatment), and the collection point was within 50 cm of the tea tree. The surface soil (0–20 cm, 3 mL/sample) of the tea root was collected. Thus, the soil samples from the 3 sampling sites of each treatment were uniformly mixed and divided into 3 samples, and Shanghai Biozeron Biotechnology Co., Ltd. (Shanghai, China) received all of the soil samples for the 16S rRNA gene sequencing assay. Microbial DNA was extracted from the soil samples using an E.Z.N.A.^®^ Soil DNA Kit (Omega Bio-Tek, Norcross, GA, USA), according to the manufacturer’s protocols. The V4-V5 region of the bacteria 16S ribosomal RNA gene was amplified by PCR, using primers 515F and 907R, where the barcode was an eight-base sequence unique to each sample. Amplicons were extracted from 2% agarose gels and purified using an AxyPrep DNA Gel Extraction Kit (Axygen Biosciences, Union City, CA, USA) according to the manufacturer’s instructions. All the PCR steps were performed by ABI GeneAmp^®^ 9700. The resulting purified amplicons were pooled in equimolar concentrations and paired-end sequenced on an Illumina MiSeq PE250 platform (Illumina, San Diego, CA, USA).

Raw fastq files were first demultiplexed using in-house perl scripts, according to the barcode sequence information for each sample, with the following criteria: (i) The 250 bp reads were truncated at any site receiving an average quality score <20 over a 10 bp sliding window, discarding the truncated reads which were shorter than 50 bp; (ii) exact barcode matching, whereby 2 nucleotides were mismatched in primer matching, and reads containing ambiguous characters were removed; (iii) only sequences that overlapped longer than 10 bp were assembled according to their overlap sequence. Reads that could not be assembled were discarded.

Sequences were clustered into operational taxonomic units (OTUs) at 97% similarity (identical) using the Deblur denoising algorithm, which removes noise due to sequencing error [32]. The alpha diversity indices were calculated with MOTHUR software at a 97% similarity based on the OTU clustering results, including the *Chao1* index, *H*, *E*, and *C* (see Section 2.3 for more information, and *Chao1* = S + F_1_^2/2F_2_, F1 and F2 are the counts of singletons and doubletons).

### 2.5. Data Analysis

A statistical analysis was conducted using GraphPad Prism 7 (GraphPad Software, Inc.) and SPSS 25.0 (IBM Corporation, Armonk, NY, USA). Two-way repeated-measures ANOVAs were used to analyze the effects of the sampling year (2020 vs. 2021), the ecological shading treatment (SS, ES, WS, SSCK and EWCK), and their interaction on the measured foliar indices of soluble nutrients and bioactive components, the quality index of tea leaves, the population occurrence of the key pests of *R. cacaonis* and *E. oblique*, and the community diversity indexes (*H*, *E*, *D,* and *C*) of insects in the tea plantation. Moreover, one-way ANOVA was used to analyze the effects of ecological-shade treatment by roadside trees on the diversity indices of the soil microbial communities. A Venn diagram was used to count the common and unique OTUs in multiple samples, and the OTU samples with a 97% similar level were selected for analysis. Principal coordinate analysis (weighted PCoA) of communities based on Bray–Curtis dissimilarity matrices across different ecological shading treatments was used. Additionally, an *LSD* test or *t* test at *p* < 0.05 was used to analyze the significant differences between/among treatments.

## 3. Results

### 3.1. Effects of Roadside Trees Ecological Shading on the Foliar Soluble Nutrients of Tea Plants

Three vital foliar soluble nutrients (i.e., soluble sugars, soluble proteins, and free fatty acids) were measured in the tea leaves under different ecological shading treatments in 2020 and 2021. The foliar contents of soluble nutrients were significantly impacted by the sampling year (*F* ≥ 4.40, *p* ≤ 0.05), and ecological shading had significant effects on the foliar content of soluble sugars (*F* = 6.04, *p* = 0.002) and free fatty acids (*F* = 7.10, *p* = 0.001), while the interaction between ecological shading and sampling year had significant effects on the contents of soluble sugars (*F* = 6.14, *p* = 0.002) and free fatty acids (*F* = 7.44, *p* = 0.001) in tea leaves (Table 1). South shading (SS) and east shading (ES) significantly reduced the foliar content of soluble sugars, when compared to the corresponding control of SSCK or EWCK (*p* < 0.05), and significantly increased the foliar content of free fatty acids (*p* < 0.05), while west shading (WS) had no significant effects on the foliar contents of soluble nutrients (*p* > 0.05; Figure 2).

Moreover, the foliar contents of soluble nutrients did not significantly differ between the two controls, SSCK and EWCK (the east-west row direction of tea plants and roadside trees of SSCK and the north-south row direction of tea plants and roadside trees of EWCK) (*p* > 0.05; Figure 2). There were no discernible differences in the foliar contents of soluble nutrients between the ecological shading treatment SS in comparison to ES and WS (*p* > 0.05). The foliar content of soluble sugars for WS was significantly higher than ES, while the free fatty acids showed the opposite trend. (Figure 2).

### 3.2. Effects of Roadside Tree Ecological Shading on the Bioactive Components of Tea Plants

The foliar caffeine and theanine contents were significantly influenced by ecological shading, sampling year, and their interaction. (*F* ≥ 3.72, *p* ≤ 0.02; Table 1); while sampling year had no significant effect on the foliar contents of polyphenols (*F* = 1.34/*p* = 0.26; Table 1). Compared with the two controls, all three ecological shading treatments significantly increased the contents of caffeine and theanine, and decreased the content of tea polyphenols (*p* < 0.05; Figure 3).

Moreover, the tea polyphenols in EWCK were significantly higher than in SSCK, and a reverse trend was seen for the theanine content of tea (*p* < 0.05; Figure 3). The foliar caffeine and theanine contents of the tea plants in SS and WS were all noticeably higher than those in ES. The foliar theanine content of tea plants in SS was significantly higher than in WS (*p* < 0.05; Figure 3).

### 3.3. Effects of Roadside Tree Ecological Shading on the Leaf Quality Indices of Tea Plants

The two indices of catechin quality index and phenol/ammonia (P/A) ratio of leaf quality were significantly influenced by ecological shading, sampling year, and their interaction. (*F* ≥ 16.89, *p* < 0.001; Table 1). The shading treatments all markedly improved the catechin quality index value and further markedly reduced the P/A ratio of tea when compared to the respective controls (*p* < 0.05; Figure 4).

The P/A ratio in the tea leaves in the EWCK control was significantly higher than that in SSCK (*p* < 0.05), and there was no significant difference in the value of the Catechin quality index between EWCK and SSCK (*p* > 0.05; Figure 4). Moreover, there were no significant differences in the P/A ratios of the tea among the three shading treatments (*p* > 0.05; Figure 4), while significant differences in the value of the Catechin quality index in tea leaves were found between each pair of the three ecological shading treatments, with a descending tread of ES > SS > WS (*p* < 0.05; Figure 4).

### 3.4. Effects of Roadside Tree Ecological Shading on Population Occurrence and Community Diversity of Insects in the Tea Plantation

#### 3.4.1. Population Occurrence of Two Key Species of Pests

The population dynamics of the two important species *R. cacaonis* and *E. oblique* were significantly influenced by ecological shading and sampling year (*F* ≥ 2.94, *p* ≤ 0.04; Table 1), while their interaction exhibited no significant effects on the population dynamics of *R. cacaonis* (*F* = 0.78, *p* = 0.55) and *E. oblique* (*F* = 0.40, *p* = 0.80; Table 1). The shading treatments (SS and ES) significantly reduced the population occurrence of *R. cacaonis* and *E. oblique*, when compared to the corresponding SSCK or EWCK controls. (*p* < 0.05; Figure 5). Moreover, there was no significant difference in the population dynamics of *R. cacaonis* or *E. oblique* between the two controls of SSCK and EWCK, or among the three ecological shading treatments (*p* > 0.05; Figure 5).

#### 3.4.2. Community Diversity of Insects

Roadside tree ecological shading significantly affected the Pielou evenness index (*E*) (*F* = 5.03, *p* = 0.006) and the Simpson dominance index (*C*) (*F* = 5.22, *p* = 0.005). There were significant differences in the values of the Shannon–Wiener index (*H*) (*F* = 53.84, *p* < 0.001), Margalef richness index (*D*) (*F* = 69.41, *p* < 0.001), and Simpson dominance index (*C*) (*F* = 54.68, *p* < 0.001) of the insect community between the two sampling years of 2020 and 2021. Additionally, there was a significant interaction between the sampling year and ecological shading, which had an impact on the insect community’s Simpson dominance index (*C*). (*F* = 4.38, *p* = 0.01; Table 1).

The ecological shading treatments significantly increased the values of the Pielou evenness index (*E*) and the Simpson dominance index (*C*), when compared to the corresponding SSCK or EWCK control. ES also significantly increased the values of the Simpson dominance index (*C*) (*p* < 0.05; Figure 6). Additionally, in the ecological shading treatment WS, the Margalef richness index (*D*) was significantly higher, compared to SS (*p* < 0.05). The insect community of ES had a Simpson dominance index (*C*) value that was significantly higher than that of SS and WS (*p* < 0.05; Figure 6). Between the two controls, there were no appreciable differences in the values of any of the insect community diversity indices (*p* > 0.05; Figure 6).

### 3.5. Effects of Roadside Tree Ecological Shading on the Diversity of Soil Microbes

#### 3.5.1. Taxonomic Composition of Soil Microbes

A total of 3,295,597 sequences and 7965 OTUs were obtained from the 15 samples by 16S rRNA sequencing of the surface soils near roots in the three ecological shading treatments and corresponding controls (3 soil samples from 3 sampling sites of each treatment). All the soil microbes belonged to 37 phyla and 94 classes (Figure 7a). The majority of OUTs (79.00%) belonged to the nine phyla, i.e., *Proteobacteria*, *Actinobacteriota*, *Acidobacteriota*, *Chloroflexi*, *Myxococcota*, *Bacteroidota*, *Gemmatimonadota*, *Verrucomicrobiota*, and *Patescibacteria* (Figure 7a). As shown in Figure 7a, the shading treatment of ES significantly increased the relative abundance of *Acinetobacter* in the soil and significantly decreased the relative abundance of bacteria *Chloroflexi*. While shading increased the abundance of *Proteobacteria*, *Acidobacteriota*, and so on, but not significantly.

The Venn diagram demonstrated that there were 1805 common OUTs in all the samples, accounting for 22.66% of the total composition of the soil microbes, and the ecological shading treatment ES had the highest number of unique OUTs (i.e., 684), while WS had the lowest number of specific OUTs (i.e., 276), accounting for 8.58% and 3.46% of the total composition of the soil microbes, respectively (Figure 7b). Moreover, the number of common OTUs (SS and control of SSCK) was 3419, accounting for 42.90% of the total; while for ES and the control EWCK, this was 2836, and for WS and the control EWCK, it was 2828, accounting for 35.60% and 35.50% of the total composition of the soil microbes, respectively (Figure 7b). The PCoA results (Figure 7c) showed that the communities of microbes clustered strongly in different ecological shading treatments, and the corresponding controls. Specifically, the first coordinate (PCoA1) separated the ecological shading treatments of ES and SS from the other ecological shading treatments WS, SSCK, and EWCK, while the second coordinate (PCoA2) explained the remaining 15.15% of the discrepancy.

#### 3.5.2. Community Diversity of Soil Microbes

The community diversity of microbes in the tea plantation soil could be assessed by computing the values of several biodiversity indexes (including *Chao1*, Shannon (*H*), Pielou evenness (*E*), and Simpson dominance (*C*) index). The outcomes clearly showed that the ecological shading treatments, as well as the controls of SSCK and EWCK, did not significantly differ in their values of the Pielou evenness index (*F* = 1.16, *p* = 0.38) and Simpson dominance index (*F* = 1.87, *p* = 0.19) of the soil microbial community (Table 2). While the ES shading, *Chao1* index, and Shannon index of soil microorganisms were significantly increased (*p* < 0.05, Table 2). Combining the results of Figure 7, it can be observed that the shading treatment changed the soil microbial diversity to a certain extent.

## 4. Discussion

### 4.1. Effects of Roadside Tree Ecological Shading on the Foliar Nutrients, Bioactive Compounds, and Quality Indexes of Tea

Shading practices can reduce the heat stress on plant growth, by intercepting radiation from the sun, and the natural shading of crops by trees is ideal but quite challenging [20]. Ecological shading by roadside trees is a special shading method in tea plantations, and here, for the first time, we investigated the influence of roadside tree ecological shading on the soluble nutrients and bioactive compounds, population occurrence of key pests, and the community diversity of insects and microbes. Shade management in tea plantations is a general approach for augmenting the quality-related metabolite contents in pre-harvest tea [23,33]. However, prolonged shading may result in a decreased tea biomass, which in turn affects the synthesis of quality indices and is considered non-conducive to tea production [23,34]. In fresh tea, the colored compounds are mainly attributed to chlorophyll. The xhlorophyll content significantly decreased under dark conditions [35], while it was accumulated after a moderate shading treatment [36]. Plant growth depends on chlorophyll to fix light energy, and the rate of chlorophyll synthesis is elevated under mild shade, which compensates for the biosynthesis and affects the color of the tea soup [37,38]. Well-managed shading after the advent of a new tea flush can improve the quality of the new shoots and leaves of the tea plants [39,40]. The ecological shading provided by roadside trees is significantly different from the artificial-net shading used for tea production, and is described as more sustainable and effective. Ecological shading provides long-term intermittent shading with daily fluctuations in the angle of the sun, which is maintained over the whole period of tea production. Among the various ecological shading treatments, south shading (SS) was more prominent throughout the whole day; while in the morning and afternoon, ES and WS predominated, respectively. In this study, shading treatment significantly decreased the amount of soluble sugars and further significantly increased the amount of fatty acids in the tea leaves, when compared to the corresponding controls SSCK and EWCK. This result may have been related to the diminished light intensity on the tea under shading by the roadside trees (see Figure 2).

There are abundant bioactive compounds of great importance in tea leaves, e.g., polyphenols, caffeine, theanine, and catechins, etc., which have significant impacts on the tea quality, contributing to its distinct taste and aroma [33,41,42,43,44]. Theanine is a unique nonproteinogenic amino acid that elicits an umami taste and greatly affects tea quality [45]. It also has several health benefits, such as lowering blood pressure, and relieving stress and anxiety, etc. [46]. Caffeine, a stimulant of the central nervous system, is widely used in food, beverages, and medicine. Many studies have proven that tea polyphenols have high antioxidant activity, and many studies have proven that they are of paramount importance in reducing the risks of cardiovascular diseases, cancer, and obesity. Quality indicators of green tea, such as the catechin quality index and (P/A) ratio, have an inverse relationship. The lower the P/A ratio, the higher the catechin quality index, which will ultimately enhance the overall quality of green tea [31,47,48]. Numerous studies have demonstrated that shading can increase the foliar content of theanine, because shade-grown plants assimilate more nitrogen and theanine breaks down less [39,48,49,50]. In summer afternoons, the high temperature and strong light in tea plantations may lead to the photoinhibition of tea [51]. Shading not only mitigates photoinhibition but also reduces the local temperature, which is conducive to the synthesis of vital tea quality functional components, including theanine, EC, EGC, and EGCG [23,48,51,52,53]. However, polyphenols and their astringency can be mitigated by the elevated concentrations that make green tea more favorable [54,55]. The theanine, caffeine, and catechin content in the foliar tissues were found to be significantly higher in this study, due to the ecological shading provided by roadside trees. Additionally, this significantly improved the tea quality index, decreased the phenol ammonia (P/A) ratio in tea leaves, and decreased the foliar content of polyphenols, all of which was consistent with earlier studies [39,56], and this is essential for maintaining the quality of green tea in tea production.

### 4.2. Effects of Roadside Tree Ecological Shading on the Population Dynamics and Community Diversity of Insects in Tea Plantations

Shading by trees has multiple effects on tea plant growth, and it has been reported to lessen the occurrence of insect pests in tea plantations [40,56,57,58,59]. Roadside trees provide a favorable environment for the intensive production of tea plants and also preserve the natural enemies of tea pests, which will increase insect diversity. Tall trees provided a habitat for birds and promote plant abundance, with control of the population of insect pests [60]. Many insect pests occur more frequently and seriously in single planting areas of crop plants (including tea plants) [61,62,63], and the damage caused by insect pests was reduced in compound tea plantations [64]. In this study, the field investigation showed that the two key species of insect pest in the local tea plantation were *R. cacaonis* and *E. oblique*, and the ecological shading by roadside trees reduced the population abundances of *R. cacaonis* and *E. oblique* by different degrees. Moreover, the insect diversity index of the tea garden was improved to a certain extent, which was consistent with the results of previous studies. In 3.1, the shading resulted in a decreased nutrient accumulation in tea leaves, which was not conducive to the feeding and utilization of pests. As ecological shelter from roadside trees increases the content of caffeine in the leaves, we believe that the infestation with pests will be reduced, as a natural pest deterrent [65]. Here, it should be noted that in the survey in October 2020, the concentration of insects resulted in the Simpson dominance index (*C*) of ES being significantly higher than that of others. The concentration of insects was a common phenomenon in the field, and we were inclined to attribute this result as being caused by accidental factors. On the whole, the existence of roadside trees reduced the pests of tea, which was beneficial to the ecology of the tea plantations. Finally, the characteristics of the camphor tree should be mentioned. Its secretion, camphor, has an antibacterial and deworming effect and is widely used by people [66]. Combined with its tall and lush tree type and the effects on the improvement of tea quality, the camphor tree may be a good choice for tea plantation roadside trees. Further experiments need to be performed to optimize the tree species.

### 4.3. Effects of Roadside Tree Ecological Shading on the Soil Microbial Diversity in Tea Plantations

Soil microbe community diversity is an important indicator for evaluating soil health and it is closely related to crop growth [67]. Soil microbes decompose organic matter and provide plants with various nutrient elements that can be directly absorbed [68]. Studies have shown that the soil microbial diversity is higher in compound environments, where a variety of plants coexist, and a suitable soil microbial community is helpful for sustainable tea production [69,70]. Shading can significantly affect the soil microbial community and change the relative abundance of microorganisms [21,71]. Deciduous trees with high, spreading crowns drop their leaves at the end of the growing season, and these droppings and animal discharges can provide inputs for the soil ecosystem in the shadow of deciduous trees, which will facilitate the reproduction of different microorganisms. In this study, ES increased the relative abundance of microorganisms in the rhizosphere of tea plants and increased the diversity index of soil microbes, which was beneficial for sustainable soil production, without adverse effects from other treatments. It was found that the ecological shading treatment with roadside trees was beneficial or harmless to the soil health around the roots of the tea plants.

## 5. Conclusions

The ecological shading rendered by roadside trees reduced the accumulation of soluble nutrients in the tea leaves, changed the foliar content of bioactive components of tea plants, and improved the quality index of the tea, which was beneficial to the quality improvement of green tea. Moreover, the ecological shading by roadside trees reduced the population occurrence of two key species of insect pest in the tea plantation. Although it did not improve the diversity and richness of the insect community in the tea plantation, it improved the evenness of the insect community. In addition, the ecological shading by roadside trees improved the community diversity of the soil microbes. The effects of the different shading directions are shown in Appendix B. In summary, roadside trees were planted to provide shade in the tea plantation, which was beneficial to the control of insect pests and to the improvement of tea quality, as well as the biodiversity enhancement of the soil microbe community in the tea plantation (Figure 8). This study may play a pivotal role in the ecological management and landscape design of tea plantations.

## Figures and Tables

**Figure 1 biology-11-01800-f001:**
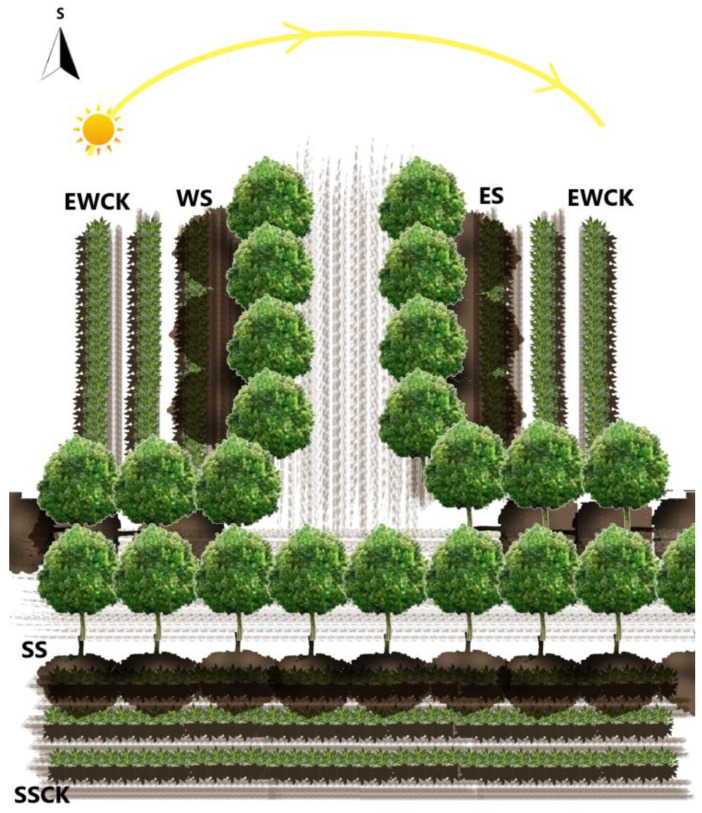
Experimental plot layout of roadside tree ecological shading (i.e., camphor trees) in the tea plantation (Note: ES—east shading; WS—west shading; EWCK—control of the shading treatments of WS and ES; SS—south shading; SSCK—control of the shading treatment of SS. The same in the following figures and tables).

**Figure 2 biology-11-01800-f002:**
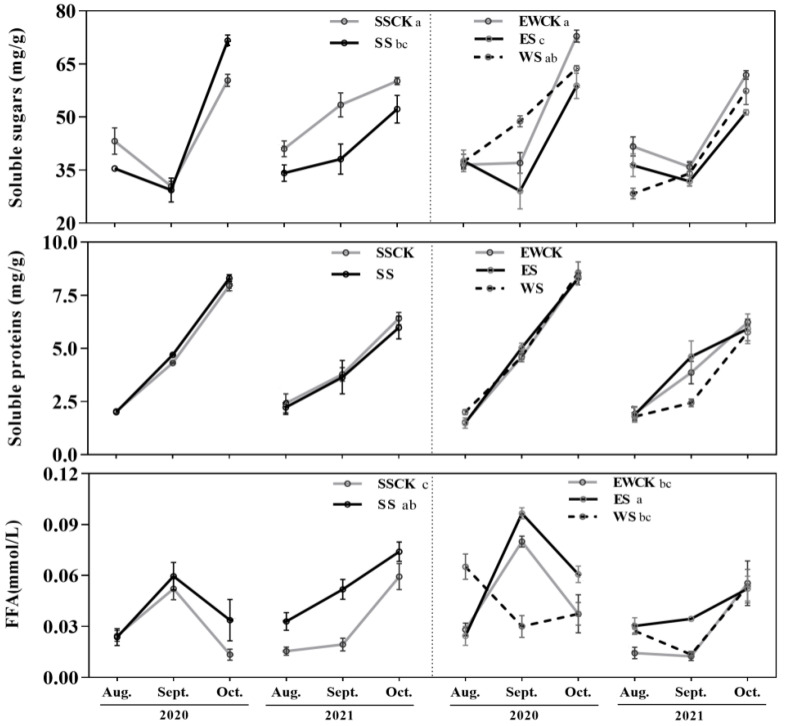
Foliar soluble nutrients of tea plants under the ecological shading treatments by roadside trees in the tea plantation (Note: different lowercase letters indicate significant differences among the ecological-shade treatments, and the control no-shading treatments, using the *LSD* test at *p* < 0.05).

**Figure 3 biology-11-01800-f003:**
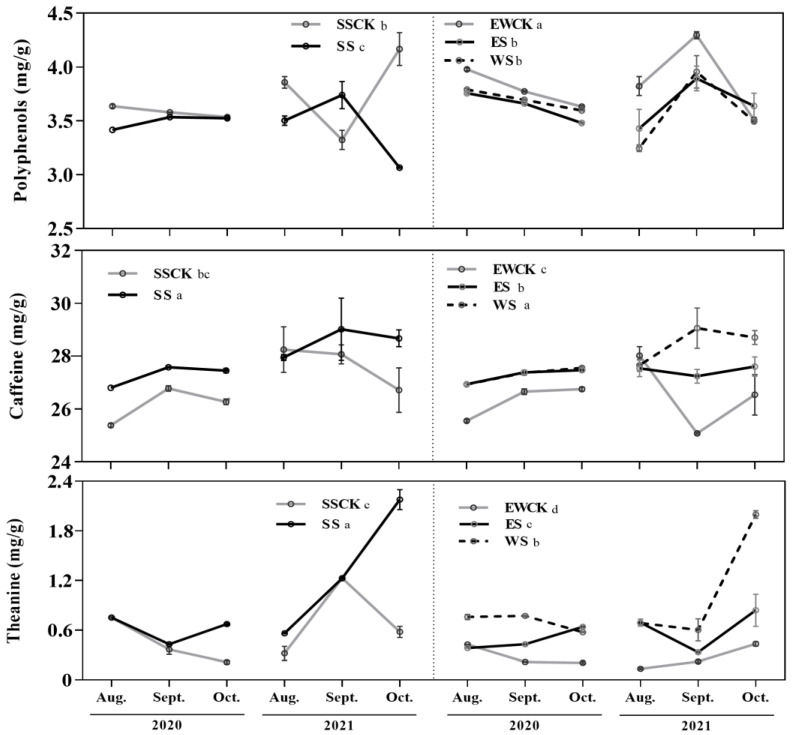
Foliar contents of bioactive components of tea plants under the roadside tree ecological shading. (Note: different lowercase letters indicate significant differences among the ecological-shade treatments, and the control no-shading treatments, using the *LSD* test at *p* < 0.05).

**Figure 4 biology-11-01800-f004:**
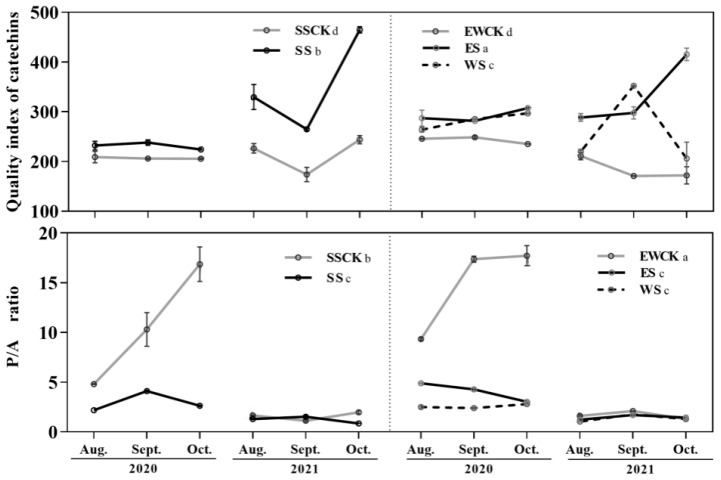
Catechin quality index and P/A ratio of tea plants under the ecological shading treatment with roadside trees in a tea plantation. (Note: different lowercase letters indicate significant differences among the ecological-shade treatments, and the control no-shading treatments, using the *LSD* test at *p* < 0.05).

**Figure 5 biology-11-01800-f005:**
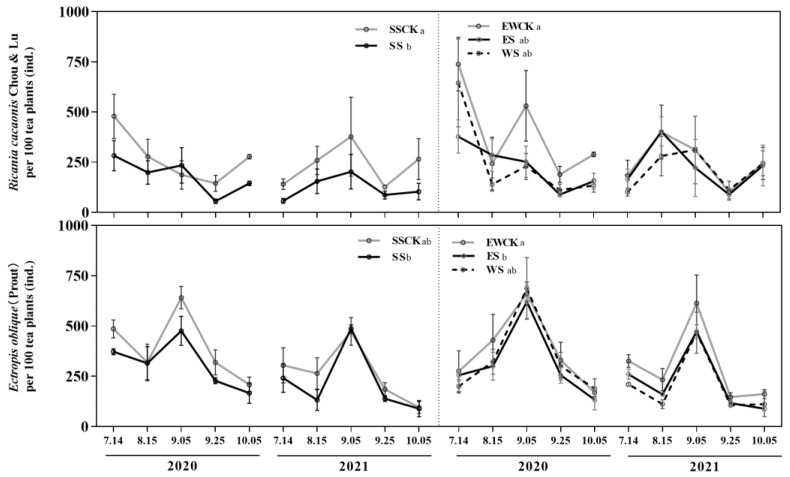
Population occurrence of two key pest species, *R*. *cacaonis* and *E. oblique,* in the ecological shading treatments with roadside trees in the tea plantation. (Note: different lowercase letters indicate significant differences among the ecological-shade treatments, and the control no-shading treatments, using the *LSD* test at *p* < 0.05).

**Figure 6 biology-11-01800-f006:**
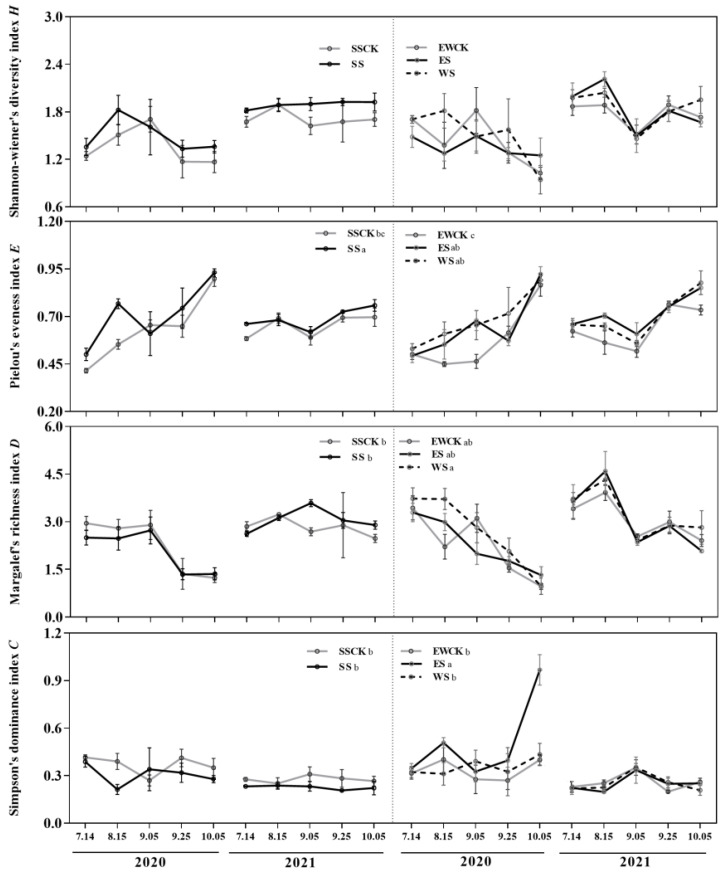
Community diversity indices of the insects that were collected in the tea plantation roadside trees used for ecological shading. (Note: different lowercase letters indicate significant differences among the ecological-shade treatments, and the control no-shading treatments, using the *LSD* test at *p* < 0.05).

**Figure 7 biology-11-01800-f007:**
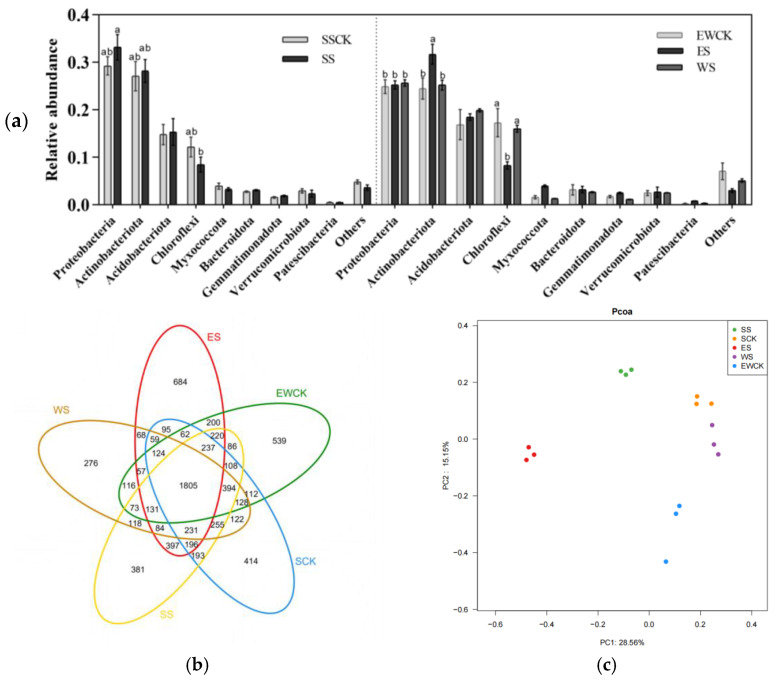
(**a**) Soil microbe relative abundances at different phylum levels of roadside trees for ecological shading (Note: Different lowercase letters indicated significantly differences among shading treatments, and the controls of SSCK and EWCK using the *LSD* test at *p* < 0.05. Bars represent standard errors (*n* = 3); (**b**) Venn diagram of soil microorganisms of the tea plantation under roadside tree ecological shading (Note: Different colors represent different samples from different ecological shading treatments. Areas where two circles of different colors overlap and marked with the number 100 mean that both samples had 100 of the same sequences with the same OTUs); (**c**) Principal coordinate analysis (i.e., PCoA) of soil microbial communities based on the Bray–Curtis dissimilarity matrices across different ecological shading treatments (Note: The percentage variation explained by the plotted principal coordinates is indicated on the axes).

**Figure 8 biology-11-01800-f008:**
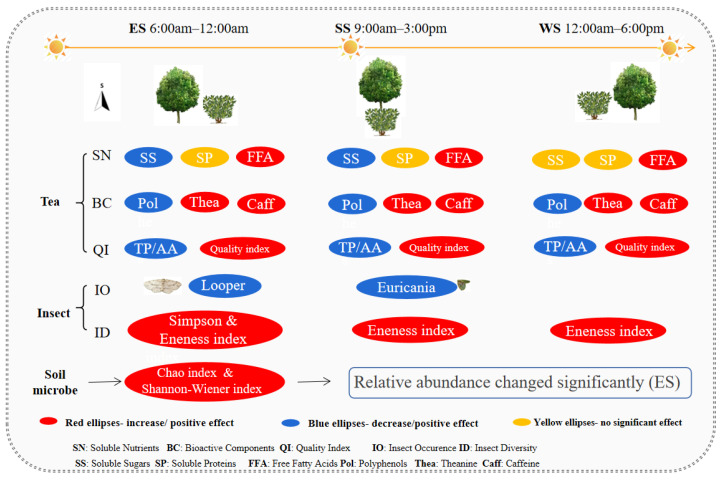
Effects of ecological shading of roadside trees on the soluble nutrients and bioactive components, community diversity of insects, and the soil microbes, as well as on the tea.

**Table 1 biology-11-01800-t001:** Two-way repeated-measures ANOVAs of ecological shading by roadside trees (S), sampling year (Y), and their interaction on the foliar contents of soluble nutrients, bioactive components, and leaf quality indices of tea plants, as well as the population occurrence of two key pests, *R. cacaonis* and *E. oblique*, and the community indices of collected insects in the tea plantation (values were *F*/*p*) (sampling time was repeated measures; * *p* < 0.05; ** *p* < 0.01; *** *p* < 0.001).

Measured Indexes		Ecological Shading (S)	Sampling Years (Y)	S×Y
Foliar soluble nutrients	Soluble sugars (mg/g)	6.04/0.002 **	4.40/0.049 *	6.14/0.002 **
	Soluble proteins (mg/g)	0.91/0.48	61.91/<0.001 ***	2.47/0.08
	Free fatty acids (mmol/L)	7.10/0.001 **	10.13/0.005 **	7.44/0.001 **
Foliar bioactive components	Polyphenols (mg/g)	34.14/<0.001 ***	1.34/0.26	7.76/<0.001 ***
	Caffeine (mg/g)	15.36/<0.001 ***	37.06/<0.001 ***	3.72/0.02 *
	Theanine (μg/g)	159.03/<0.001 ***	216.95/<0.001 ***	37.36<0.001 ***
Leaf quality	Catechin quality index	88.73/<0.001 ***	16.89/0.001 **	49.3/<0.001 ***
	P/A ratio	103.85/<0.001 ***	508.74/<0.001 ***	92.39/<0.001 ***
Population dynamics of two key species	*Ricania cacaonis*	4.09/0.01 *	5.02/0.04 *	0.78/0.55
	*Ectropis oblique*	2.94/0.04 *	32.64/<0.001 ***	0.40/0.80
Community diversity of insects	*H*	1.20/0.34	53.84/<0.001 ***	0.27/0.90
	*E*	5.03/0.006 **	3.86/0.063	1.25/0.32
	*D*	2.57/0.07	69.41/<0.001 ***	0.73/0.58
	*C*	5.22/0.005 **	54.68/<0.001 ***	4.38/0.01 *

**Table 2 biology-11-01800-t002:** Community diversity indices of soil microbes in a tea plantation that was ecologically shaded by roadside trees.

Diversity Indices	SSCK	SS	EWCK	ES	WS	*F*/*p*
*Chao1*	4086.3 ± 352.2 a	4477.9 ± 72.2 a	3448.0 ± 315.7 b	4126.7 ± 361.8 a	3356.8 ± 318.5 b	7.48/0.005 **
*H*	9.95 ± 0.15 ab	10.12 ± 0.14 a	9.71 ± 0.26 b	10.11 ± 0.10 a	9.69 ± 0.06 b	5.42/0.01 *
*E*	0.8449 ± 0.0038	0.8539 ± 0.0128	0.8423 ± 0.0181	0.8557 ± 0.0033	0.8428 ± 0.0036	1.16/0.38
*C*	0.0027 ± 0.0004	0.0025 ± 0.0004	0.0034 ± 0.0011	0.0023 ± 0.0002	0.0030 ± 0.0002	1.87/0.19

Note: roadside trees and ecological shading had an impact on the diversity indices of soil microbes, which were examined using a one-way ANOVA; different lowercase letters signify significant differences in SS, ES, and WS, and the controls SSCK and EWCK ecological shading treatments using an *LSD* test at *p* < 0.05, * *p* < 0.05; ** *p* < 0.01.

## Data Availability

The data obtained in this study are presented “as is” in at least one of the figures or tables embedded in the manuscript.

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
