# Peer review of "Impacts of Ecological Shading by Roadside Trees on Tea Foliar Nutritional and Bioactive Components, Community Diversity of Insects and Soil Microbes in Tea Plantation"

_biology, 2022, doi:10.3390/biology11121800_

Round 1
Reviewer 1 Report
The manuscript is nicely written and presented in a very favourable way.
Author Response
Thank you for your review and approval. Best wishes for you and your family. Thank you very much.
Reviewer 2 Report
Dear authors,
The manuscript by Zou et al. is describing the impacts of ecological shading by roadside trees on tea foliar nuritional and bioactive components, community diversity of insects and soil microbes in tea plantation.
The manuscript is well-written and clearly presented, the used methods are appropriate, the results are very convincing, I warmly recommend it for publication in Biology journal.
Comment 1: Page 37 Line 31 shading instead of shanding
Author Response
Thank you for your review and approval. I have corrected the spelling mistakes in the article. Best wishes to you and your family. Thank you very much.
Reviewer 3 Report
Thank you for allowing me to review the manuscript on the Impacts of Ecological Shading by Roadside Trees on Tea Foliar Nutritional and Bioactive Components, Community Diversity of Insects and Soil Microbes in Tea Plantation.
The cultivation of tea is an economically important activity and therefore further research into optimising growth conditions is always welcome.
This article is well-written and presents the benefits of shade from Roadside trees on tea plants. It was also a good idea to measure several years to reduce variability in results.
That being said there are a couple of points I would like to clarify in the manuscript.
1. Can the authors quantify the benefits from shading in the abstract ?
2. What distance from the shade plants were the soil samples taken?
3. Did the authors factor in the properties of Camphor trees into their discussion ?
By this I mean that Camphor is insecticidal, antimicrobial, antiviral, anticoccidial, anti-nociceptive and potentially toxic to some species.
Camphor—A Fumigant during the Black Death and a Coveted Fragrant Wood in Ancient Egypt and Babylon—A Review. Weiyang Chen, Ilze Vermaak, and Alvaro Viljoen*Molecules. 2013 May; 18(5): 5434–5454. Published online 2013 May 10. doi: 10.3390/molecules18055434 PMCID: PMC6270224 PMID: 23666009
4. Line 442, the authors state that In summer afternoons, the elevated temperature and light intensity in tea plantations caused light inhibition in the middle and lower belts of the Yangtze River Basin [51]. When I read this reference, the article was mostly about the effects of shade from maize plants on tea in the same plantation. However I did note that reference 32 from this manuscript studied the “ Effects of Light Intensity and Temperature under Shading Treatments on the Metabolites in Tea”. PLoS ONE 2014, 9, e112572. Was this a more appropriate reference ?
5. Reference 51 covers a very similar subject to the effect of roadside tree shading. Zou, Y.; Shen, F.; Zhong, Y.; Lv, C.; Pokharel. S.S.; Fang, W.; Chen, F. Impacts of Intercropped Maize Ecological Shading on Tea Foliar and Functional Components, Insect Pest Diversity and Soil Microbes. Plants (Basel). 2022, 11, 1883. https://doi.org/10.3390/plants11141883. Since this study also looked at the effects of shade on tea plants in the same manner as this manuscript I wondered why they authors did not compare results a lot more frequently?
6. The period of the previous study on benefits of shading from maize intercropping (ref 51) covered the period 2019 and 2020, while the study of Roadside trees covered the period 2020-2021 at the same experimental tea plantation. Were the 2020 sampling sites from the maize study separate from 2020 sampling sites for the Roadside tree study?
7. I asked the previous question because I noted that some of the control values in Fig 2, Fig 3, Fig 4 seem similar in the Maize intercropping study and the Tree Line study ? Would it not be useful to mention that some of this data has been used before ?
8. Can the authors quantify the benefits from shading in the abstract ?
FFigure 2,
Bottom panel (B)1/1000 fold difference in measurements between previous maize study and Line of trees study ? is something wrong with units ? Why is there a continuation line between Oct and Aug of 2020-2021 ? This is a 9 month gap, this should be accounted for in the graph.
Figure 3,
Something wrong with units on Theanine. 1/1000 difference in scale from maize shading to tree shading study? Why is there a continuation line between Oct and Aug of 2020-2021 ? This is a 9 month gap, this should be accounted for in the graph.
Figure 4, Why is there a continuation line between Oct and Aug of 2020-2021 ? This is a 9 month gap, this should be accounted for in the graph.
Typos
Line 58. Could you substitute “predatory natural enemies” for natural predatory species ?
Line 74. I would consider starting a new paragraph here
Line 120. What is straight lighting time ?
Line 142. Amino group à amino terminus
Line 153. By using à using
Line 206. Volume of soil collected in sampling ?
Line 341. “the value of the Margalef”, extra space between words !
Line 418. the colour of tea soup ?
Line 451 Reference ?
Line 498. Error
Line 519. Should be at least one.
